# Targeted Chromosomal Barcoding Establishes Direct Genotype-Phenotype Associations for Antibiotic Resistance in *Mycobacterium abscessus*

Juan Calvet-Seral,[a,b] Estefanía Crespo-Yuste,[a,b] Vanessa Mathys,[c] Hector Rodriguez-Villalobos,[d] Pieter-Jan Ceyssens,[c] Anandi Martin,[e,f] Jesús Gonzalo-Asensio[a,b,g]

[a]Grupo de Genética de Micobacterias, Departamento de Microbiología, Facultad de Medicina, Universidad de Zaragoza IIS-Aragón, Zaragoza, Spain
[b]CIBER Enfermedades Respiratorias, Instituto de Salud Carlos III, Madrid, Spain
[c]Unit of Human Bacterial Diseases, Sciensano, Brussels, Belgium
[d]Cliniques Universitaires Saint-Luc, Microbiology Department, Université Catholique de Louvain, Brussels, Belgium
[e]Institute of Experimental and Clinical Research, Université Catholique de Louvain, Woluwe-Saint-Lambert, Belgium
[f]Syngulon, Seraing, Belgium
[g]Instituto de Biocomputación y Física de Sistemas Complejos, Zaragoza, Spain

**ABSTRACT** A bedaquiline-resistant *Mycobacterium abscessus* isolate was sequenced, and a candidate mutation in the *atpE* gene was identified as responsible for the antibiotic resistance phenotype. To establish a direct genotype-phenotype relationship of this mutation which results in a Asp-to-Ala change at position 29 (D29A), we developed a recombineering-based method consisting of the specific replacement of the desired mutation in the bacterial chromosome. As surrogate bacteria, we used two *M. abscessus* bedaquiline-susceptible strains: ATCC 19977 and the SL541 clinical isolate. The allelic exchange substrates used in recombineering carried either the sole D29A mutation or a genetic barcode of silent mutations in codons flanking the D29A mutation. After selection of bedaquiline-resistant *M. abscessus* colonies transformed with both substrates, we obtained equivalent numbers of recombinants. These resistant colonies were analyzed by allele-specific PCR and Sanger sequencing, and we demonstrated that the presence of the genetic barcode was linked to the targeted incorporation of the desired mutation in its chromosomal location. All recombinants displayed the same MIC to bedaquiline as the original isolate, from which the D29A mutation was identified. Finally, to demonstrate the broad applicability of this method, we confirmed the association of bedaquiline resistance with the *atpE* A64P mutation in analysis performed in independent *M. abscessus* strains and by independent researchers.

**IMPORTANCE** Antimicrobial resistance (AMR) threatens the effective prevention and treatment of an ever-increasing range of infections caused by microorganisms. On the other hand, infections caused by *Mycobacterium abscessus* affect people with chronic lung diseases, and their incidence has grown alarmingly in recent years. Further, these bacteria are known to easily develop AMR to the few therapeutic options available, making their treatment long-lasting and challenging. The recent introduction of new antibiotics against *M. abscessus*, such as bedaquiline, makes us anticipate a future when a plethora of antibiotic-resistant strains will be isolated and sequenced. However, in the era of whole-genome sequencing, one of the challenges is to unequivocally assign a biological function to each identified polymorphism. Thus, in this study, we developed a fast, robust, and reliable method to assign genotype-phenotype associations for putative antibiotic-resistant polymorphisms in *M. abscessus*.

**KEYWORDS** bedaquiline, Bq, Bdq, drug resistance, nontuberculous mycobacteria, cystic fibrosis, chronic obstructive disease, barcoding, recombineering

Address correspondence to Jesús Gonzalo-Asensio, jagonzal@unizar.es.

The authors declare no conflict of interest.

10.1128/spectrum.05344-22 **1**

The number of pulmonary infections caused by nontuberculous mycobacteria (NTM) have increased worldwide in the last decades (1, 2). In 1987, the U.S. Centers for Disease Control and Prevention (CDC) estimated the NTM disease rate at 1.8/100,000 persons (3). Data from North American studies between 2006 and 2012 suggested a disease rate of 5 to 10 per 100,000 persons, demonstrating its rising incidence in recent years (4). Of those NTM diseases, the major cause of infections by fast-growing mycobacteria was species from the *Mycobacterium abscessus* complex, especially in patients with chronic lung diseases, such as cystic fibrosis or chronic obstructive disease (5, 6). The *M. abscessus* complex is currently divided into three different subspecies (*M. abscessus*, *M. massiliense*, and *M. bolletii*) in which *M. abscessus* subsp. *abscessus* (here, *M. abscessus*) is the most common pathogen (7). Similarly to multidrug-resistant (MDR) tuberculosis, *M. abscessus* lung diseases are difficult to treat due to limited therapeutic options and its intrinsic and easy development of drug resistance (8), with an antimicrobial treatment success rate of less than 50% (9, 10).

Different health institutions recommend the use of a combination of oral macrolides (clarithromycin or azithromycin) with a parenteral $\beta$-lactam (cefoxitin or imipenem) and an aminoglycoside (amikacin) for a period of 6 to 12 months that can be extended to even 2 years (5). These long-term treatments are usually poorly tolerated by patients, due to the manifestation of side effects and drug-related toxicity against the different drugs used (10, 11). In addition, *M. abscessus* can develop an inducible macrolide resistance conferred by the ribosomal methyltransferase gene *erm*(41) or acquire resistance due to mutations in the 23S rRNA gene *rrl* (12). This limited use of oral drugs currently makes essential the use of injectable antimicrobials. Therefore, there is a need to study new strategies to combat *M. abscessus* infections, especially for macrolide-resistant strains. Several studies have evaluated the use of antimicrobials used in different mycobacterial infections, like clofazimine (13), which is used to treat *M. leprae* and MDR tuberculosis, and bedaquiline (14), the first drug approved by the Food and Drug Administration in 40 years for treatment of MDR tuberculosis.

Bedaquiline is a diarylquinoline that inhibits the mycobacterial ATP synthase by targeting the subunit C of the F0/V0 complex, preventing the rotor ring from acting as ion shuttle (15). Sequencing of spontaneous resistant mutants obtained in the laboratory revealed that mutations in the coding gene of the ATPase subunit C, the *atpE* gene, are responsible for acquired resistance against bedaquiline in different mycobacterial species (16).

Emergence and spread of antimicrobial resistance is one of the top 10 global public health threats declared by the WHO (17). As new mutations are daily identified in drug-resistant isolates, mainly thanks to lowering costs in the implementation of whole-genome sequencing technology, characterization of these new mutations is essential to understand their impact in bacterial biology, and specifically to corroborate their roles in antimicrobial resistance. This would allow confirmation of the genetic-phenotype linkage of candidate mutations and discarding other mechanisms like drug tolerance, based on phenotypic tolerability to drugs independent of a genetic association (18).

A classic strategy commonly used to confirm the genetic-phenotype association of a mutant resistant allele is the complementation and overexpression of the mutant allele in a sensitive strain with a plasmid. However, the resultant merodiploid strain carries both mutant and sensitive alleles, which might lead to interfering phenotypes. In addition, overexpression of a gene product at nonphysiological levels could also lead to exacerbated phenotypes which do not reflect a clinical situation. For example, the F0/V0 complex of bacterial ATPase is composed of several C subunits in which, if one of the subunits is susceptible to being blocked by bedaquiline, the whole rotor ring can be blocked. This case has been observed when trying to confirm two *atpE* mutations in *M. abscessus* (with an Asp-to-Ala change at position 29 [D29A] and or A64P), in which expression of these mutant alleles of the *atpE* gene in a multicopy plasmid did not increase the MIC values of bedaquiline. However, when selective pressure led to a

spontaneous recombination replacing the whole sensitive allele in the *M. abscessus* chromosome, the resultant bacteria increased the MIC 256 times (19).

Thus, a more optimal strategy to confirm genetic-phenotype associations in antimicrobial resistance, or whatever biological trait is to be interrogated, implies genetic manipulations directly in the bacterial chromosome. However, genetic tools for mycobacteria are limited, in comparison with widely used genetic model microorganisms such as *Escherichia coli*; and genetic manipulation of *M. abscessus* is even more restricted. For example, some genome mutagenesis systems that work in other species of mycobacteria, like the thermosensitive-counterselective *sacB* system and the mycobacteriophage system, do not work in *M. abscessus* (20). In the last years, suicide plasmid strategies based on counterselection with the *M. tuberculosis katG* sensitive allele or the *E. coli galK* gene have been successfully used to generate different *M. abscessus* knockouts (21–24), even if construction of mutant strains hardly correlated with antimicrobial resistance.

A recombineering system is one of the most used methods to perform genetic manipulations in the chromosome of mycobacteria. It was first described in *Escherichia coli* in 2000 (25) and later applied to *M. smegmatis* and *M. tuberculosis* (26). It consists of the transformation of an allelic exchange substrate (AES) in a strain expressing the Gp60 and Gp61 proteins from the Che9c mycobacteriophage. Using a double-strand AES (dsAES) with long homology flanking regions (around 1 to 1.5 kbp) flanking an antibiotic resistance cassette, recombineering works with acceptable efficiency in *M. abscessus*. However, the percentage of double-crossover mutants of *M. abscessus* can be low in comparison with other mycobacteria due to the natural ability of *M. abscessus* to circularize linear dsDNA fragments (20, 27, 28). A recombineering system also allows the use of short single-stranded oligonucleotides as AES (ssAES) in mycobacteria (29); however, as far as we know, there have been no reports of the use of ssAES in *M. abscessus* to perform chromosome mutations.

In this work, we used a recombineering system to perform oligo-mediated targeted mutagenesis in the genome of two different strains of *M. abscessus*, the clinical strain SL541 and the laboratory reference strain *M. abscessus* ATCC 19977, to confirm the *atpE* D29A sequenced mutation, which is involved in resistance to bedaquiline. In addition, we demonstrated the suitability of introducing barcoding, silent mutations to enable a fast and reliable colony screening by PCR-based methodologies, avoiding the extra time needed to confirm mutations by sequencing. This barcoding methodology has been successfully used to confirm the *atpE* D29A mutation and, additionally, the *atpE* A64P mutation, which also confers bedaquiline resistance in *M. abscessus*.

## RESULTS

**Identification of a bedaquiline-resistant strain derived from an *M. abscessus* clinical isolate.** The clinical isolate *M. abscessus* SL541 was obtained from a sputum sample from a patient with cystic fibrosis. The antimicrobial resistance profile of *M. abscessus* SL541 strain was determined with Sensititre RAPMYCO2 test and showed resistance to cefoxitin, imipenem, tobramycin, ciprofloxacin, moxifloxacin, trimethoprim, sulfamethoxazole, and doxycycline (Table 1). Subsequently, a bedaquiline spontaneously resistant mutant was isolated from a single colony of *M. abscessus* SL541 plated in 7H10–albumin-dextrose-catalase (ADC) containing 4 $\mu$g/mL of bedaquiline. This clone, *M. abscessus* SL541BQR, showed a MIC against bedaquiline in broth of 8 to 4 $\mu$g/mL, whereas its parental strain, *M. abscessus* SL541, exhibited a MIC of 0.5 $\mu$g/mL. The genomic DNA of both strains was isolated and sequenced using Illumina MiSeq, identifying the A89C single-nucleotide polymorphism in the *atpE* gene. This mutation translates into the nonsynonymous D29A mutation in the AtpC subunit of the bedaquiline target ATP synthase, indicating the causing mutation of bedaquiline resistance in our *M. abscessus* SL541 strain. Accordingly, we aimed to develop a genetic strategy to establish a direct relationship between the genetic *atpE* mutation and the phenotypic resistance to bedaquiline in *M. abscessus* (Fig. 1).

**TABLE 1** Susceptibility profiles of the *M. abscessus* isolate 81327881541 (SL541) and the *M. abscessus* reference strain ATCC 19977

| Antibiotic(s) | MIC$^a$ ($\mu$g/mL) for *M. abscessus* strain: | |
| --- | --- | --- |
| | SL541 | ATCC 19977 |
| Amoxicillin-clavulanate | >64/32 | >64/32 |
| Cefoxitin | R >128 | 64 |
| Ceftriaxone | >64 | ND |
| Cefepim | >32 | ND |
| Imipenem | R >64 | 2 |
| Linezolid | ND | ≤4 |
| Amikacin | 16 | 8 |
| Tobramycin | R >16 | ND |
| Clarithromycin | 0.12 | S <1 |
| Ciprofloxacin | R >4 | ≤0.5 |
| Moxifloxacin | R >8 | 1 |
| Trimethoprim-sulfamethoxazole | R >8/152 | R >8/152 |
| Minocycline | 4 | ND |
| Doxycycline | R >16 | R >16 |
| Tigecycline | 2 | 1 |
| Bedaquiline | 0.5 | 0.062 |
| Kanamycin | 16 | 8 |

$^a$R, resistant; S, susceptible; ND, not determined.

**Specific chromosomal replacements at *atpE* codon at position 29 results in bedaquiline resistance in *M. abscessus*.** Establishing a direct genotype-phenotype link in the context of antimicrobial resistance usually requires the introduction of the candidate mutation(s) into a susceptible bacterium to reproduce the phenotypic drug resistance. However, introduction of a gene outside its original location might produce undesirable effects in the context of nonphysiological expression of the mRNA or protein products. Further, introduction of genes using plasmids has two main disadvantages: on the one hand, this strategy usually leads to merodiploid bacteria which carry wild-type and mutated copies of the gene of interest. On the other hand, plasmidic expression of genes does not resemble the copy number nor the expression levels of chromosomal genes. With this scenario, we reasoned that an optimal strategy to confirm the role of a suspected gene (or mutation) in drug resistance would require the replacement of the wild-type allele in the chromosome. This strategy has the additional advantage that chromosomal replacements would eventually lead to drug-resistant bacteria which could be selected on antibiotic-containing plates.

We used single-stranded recombineering, a technique which uses linear ssDNA substrates with flanking homology regions to direct targeted recombination in the chromosome (29). We confirmed the bedaquiline susceptibility of the *M. abscessus* SL541 strain on 7H10-ADC agar, which was in clear contrast to that of *M. abscessus* SL541BQR carrying the bedaquiline resistance mutation *atpE* D29A (Fig. 2A). We then introduced the recombineering machinery located in the pJV53 (26) plasmid into the *M. abscessus* SL541 strain. First, we assayed the kanamycin susceptibility of this strain prior to introducing the kanamycin-bearing pJV53 plasmid. Then, we selected kanamycin-resistant transformants and confirmed the presence of pJV53-positive clones by PCR amplification of a plasmid-specific region (see Fig. S1 in the supplemental material). Finally, we confirmed that introduction of the recombineering plasmid did not alter the original bedaquiline susceptibility in the *M. abscessus* SL541 pJV53 strain (Fig. 2A).

Once we constructed the *M. abscessus* recombineering strain, we induced expression of the recombineering genes, and the AES containing the desired mutation were transformed into competent bacteria. Specifically, the AES consisted of single-stranded oligonucleotides carrying either the wild-type codon (GAC, coding for Asp) or the mutated codon (GCC, coding for Ala and leading to a D29A mutation), in the central position of the ssDNA substrates. The left and right arms of the AES contained 65 and

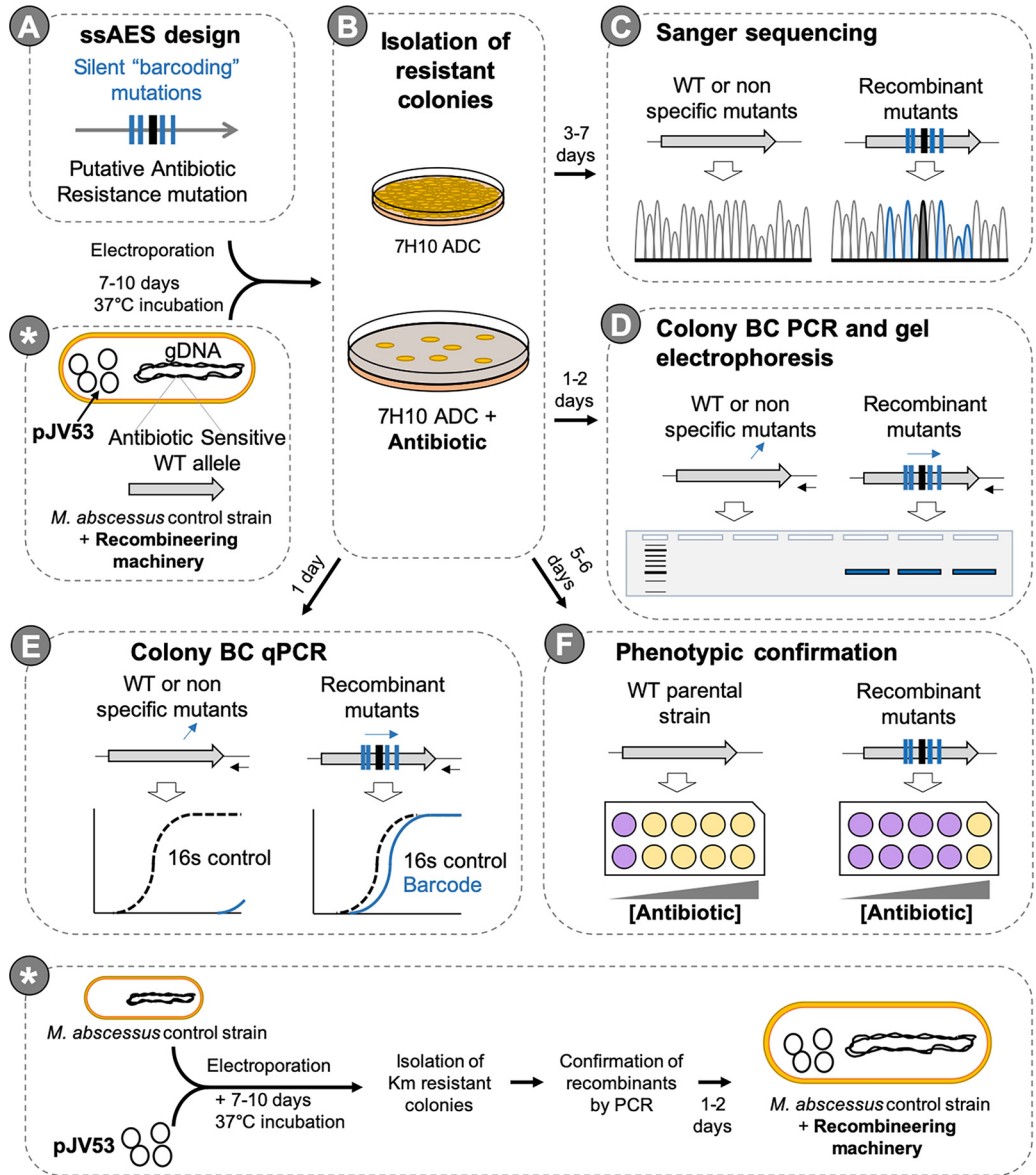

**FIG 1** Workflow scheme followed in this study. (A and B) Design of ssAES carrying silent barcoding (BC) mutations and the antibiotic resistance mutation (A) allowed the selection of resistant colonies after electroporation of *M. abscessus* carrying the recombineering machinery (B). (C to E) Sanger sequencing (C) or barcoding PCR methods (D and E) were used to confirm genotypes of recombinant colonies recovered. (F) Confirmation of the phenotype-genotype association of the mutation under study was assessed by calculating the MIC to the antibiotic. The panel marked with an asterisk indicates the experimental steps necessary to construct an *M. abscessus* recombineering strain, which added an extra 8 to 12 days to the times indicated in the figure. It is also important to consider that plasmid transformation into nonlaboratory strains can be challenging.

67 homology nucleotides which promoted the specific chromosomal replacement (Fig. 2B). Recombinant bacteria were plated in medium containing bedaquiline, to show the presence of drug-resistant colonies exclusively in bacteria transformed with the D29A allele but not in transformants with the wild-type allele (Fig. 2C). To ensure that bedaquiline-resistant bacteria arose from a similar number of transformants, we also plated bacteria without antibiotic. Results showed an equivalent number of CFU in bacteria transformed with wild-type and mutant alleles (Fig. 2C). These bedaquiline-resistant colonies were further confirmed by their increased MICs of bedaquiline relative to that for the *M. abscessus* recombineering strain (Fig. 2D). Further, in order to determine that other mutations were not contributing to our observed resistance phenotype, we sequenced the whole *atpE* gene in the recombinant colonies. We

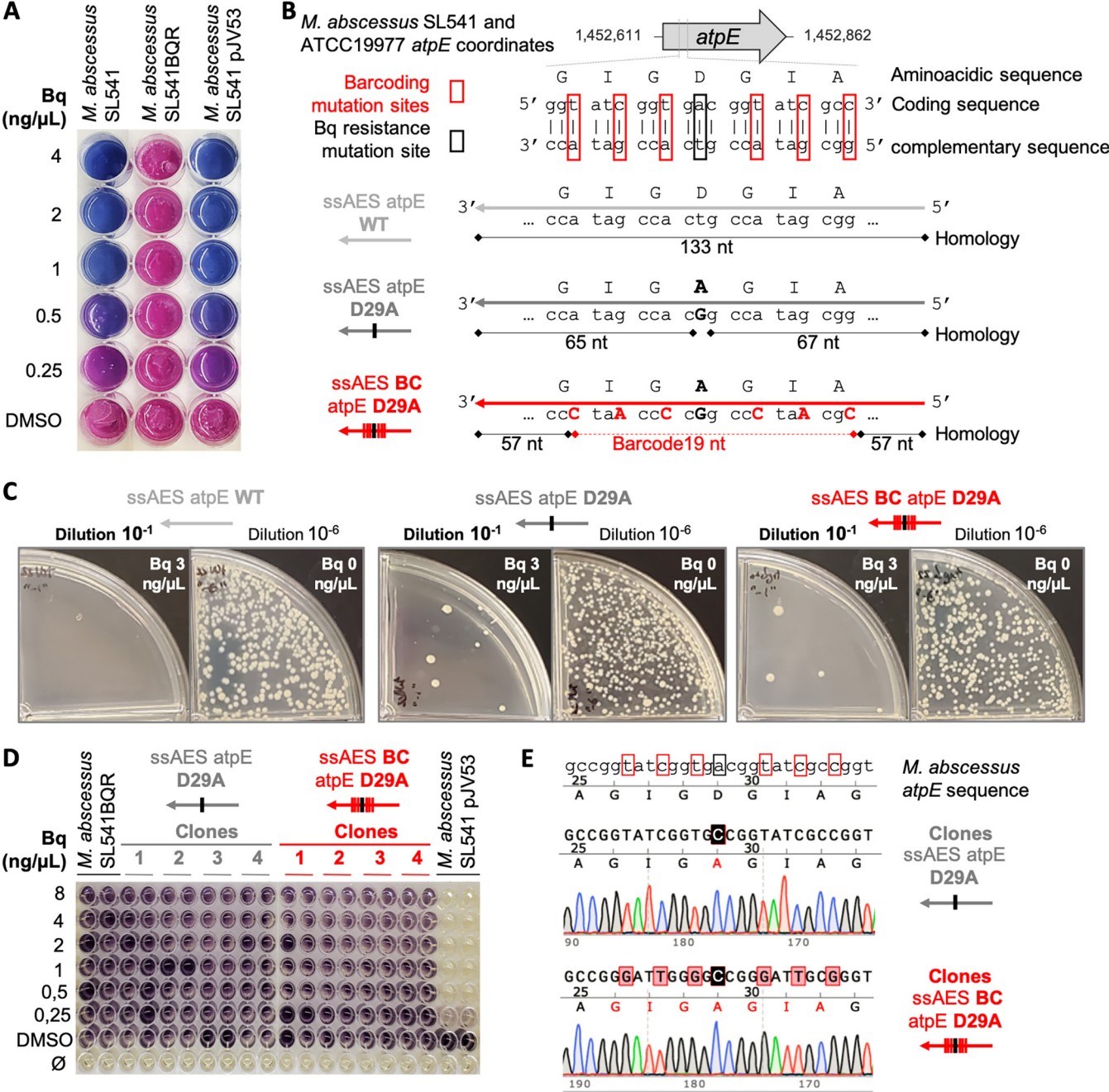

**FIG 2** Chromosomal barcoding method to establish a genotype-phenotype association for the D29A mutation in the *atpE* gene. (A) Resazurin microtiter assay in 7H10-ADC to determine MICs of bedaquiline against different *M. abscessus* SL541 strains. Wells with viable bacteria are shown in pink, whereas wells with absence of growth are shown in blue, after addition of resazurin. (B) Graphical representation of the ssAES used to mutate the 29th codon of the *atpE* gene and the alignment in *M. abscessus* SL541 and reference ATTC19977 strain genomes. (C) Electroporated bacteria with ssAES atpE wild type (WT), ssAES atpE D29A, or and ssAES BC atpE D29A were plated in the presence (dilution $10^{-1}$) or absence (dilution $10^{-6}$) of bedaquiline at 3 $\mu$g/mL. (D) MTT in 7H9Tyl-ADC was used to determine the MICs for different isolated recombinant colonies. Wells with viable bacteria are shown in dark purple, whereas wells with absence of growth are shown in yellow, after addition of MTT. (E) Sanger sequencing chromatograms of recombinant colonies of *M. abscessus* electroporated with ssAES atpE D29A (top) and ssAES BC atpE D29A (bottom).

obtained a clear and unique GCC substitution at the 29th codon position relative to the wild-type GAC codon in all analyzed colonies, indicating that this mutation was specifically responsible for the bedaquiline resistance phenotype in our recombinants (Fig. 2E; Fig. S2).

However, it should be noted that some of the *M. abscessus atpE* D29A mutants identified by this approach may have resulted from spontaneous mutations. Accordingly,

we refined our strategy to specifically detect those resistant bacteria derived from our recombineering approach. We constructed a new AES containing the GCC mutated codon, immediately flanked by three silent mutations at each side, thus not altering the coding capacity of the mutated codons (Fig. 2B). These silent modifications were aimed to act as a genetic barcode to selectively detect the incorporation of the GCC mutation by recombineering, thus ruling out spontaneous mutations that arose during laboratory manipulation. This new AES contained 94.7% identical nucleotides relative to our previous approach (Fig. 2B). Again, we obtained bedaquiline-resistant colonies by applying this strategy (Fig. 2C), which reproduced the antimicrobial resistance by elevated MICs of bedaquiline (Fig. 2D). Interestingly, once we sequenced the *atpE* gene, all bedaquiline-resistant colonies contained the GCC alanine codon flanked by the barcode silent mutations (Fig. 2E; Fig. S2). This barcoding strategy allowed us to unequivocally demonstrate that introduction of the D29A mutation at the *atpE* chromosomal location by recombineering was responsible for bedaquiline resistance in the recombinants.

**Allele-specific PCR of bedaquiline-resistant colonies allows the detection of chromosomal replacements at the *atpE* D29 codon position.** Once we demonstrated the utility of our genetic barcoding strategy to specifically detect the incorporation of the desired mutation by genomic sequencing, we optimized PCR-based methods to accelerate the whole process. First, we used allele-specific PCR, using an oligonucleotide annealing to the barcode sequence but not to the wild-type sequence, and a second oligonucleotide outside the ssDNA used as the AES (Fig. 3A and B). After PCR, using a small portion of the recombinant colonies as genetic material, we confirmed the presence of the specific 531-bp PCR band in 9/12 bedaquiline-resistant colonies transformed with the barcode oligonucleotide, but not in colonies transformed with the oligonucleotide containing solely the GCC mutation in the central position (Fig. 3C). Further, to verify that the absence of a band was not the result of a nonworking PCR, we amplified the nonrelated 16S rRNA gene as a positive control. The presence of 16S bands in all colonies analyzed indicated that lack of PCR amplification in 3/12 colonies transformed with the barcode oligonucleotide was probably the result of bedaquiline-resistant spontaneous mutations elsewhere in the chromosome. Thus, this methodology is useful to discriminate between the desired mutation and false positives due to unrelated spontaneous polymorphisms (Fig. 3C). Overall, the whole process, from the transformation of the *M. abscessus* recombineering strain with the desired mutant AES to the PCR verification of mutant colonies, requires less than two working weeks. Alternatively, we used real-time PCR to detect PCR amplification without the need for agarose gel electrophoresis. Results showed amplification of the 16S rRNA control gene (threshold cycle [$C_T$], $\approx 25$) in transformants with either the barcode or the D29A-only oligonucleotides (Fig. 3D). However, using a D29A barcode-specific oligonucleotide, we were able to specifically detect amplification in our barcode transformants ($C_T$, $\approx 25$) (Fig. 3E), in contrast to the recombinant colonies transformed with the nonbarcoded AES only containing the GCC mutation ($C_T$, $>31$) (Fig. 3D). We also confirmed the specificity of real-time PCR products after interrogation of melting curves from each amplicon (Fig. 3D and E). Using detection by real-time PCR, the entire process lasted roughly 10 days. This time expanded to roughly 20 days when the *M. abscessus* recombineering strain needed to be constructed (Fig. 1).

In another attempt to demonstrate the robustness of our method, we aimed to reproduce the techniques described here with the *M. abscessus* ATCC 19977 laboratory reference strain, carrying the pJV53 recombineering plasmid (Fig. S1), in an experiment performed by an independent researcher. We successfully reproduced results obtained with either the barcoded or the nonbarcoded oligonucleotides (Fig. S3). Altogether, our results demonstrated the robustness and reproducibility of our genetic replacement strategy and laid foundations to apply these methods in independent laboratories.

Finally, since *M. abscessus* is able to perform homologous recombination *per se*, we wanted to discard that allelic exchange of the AES occurred even in the absence of the

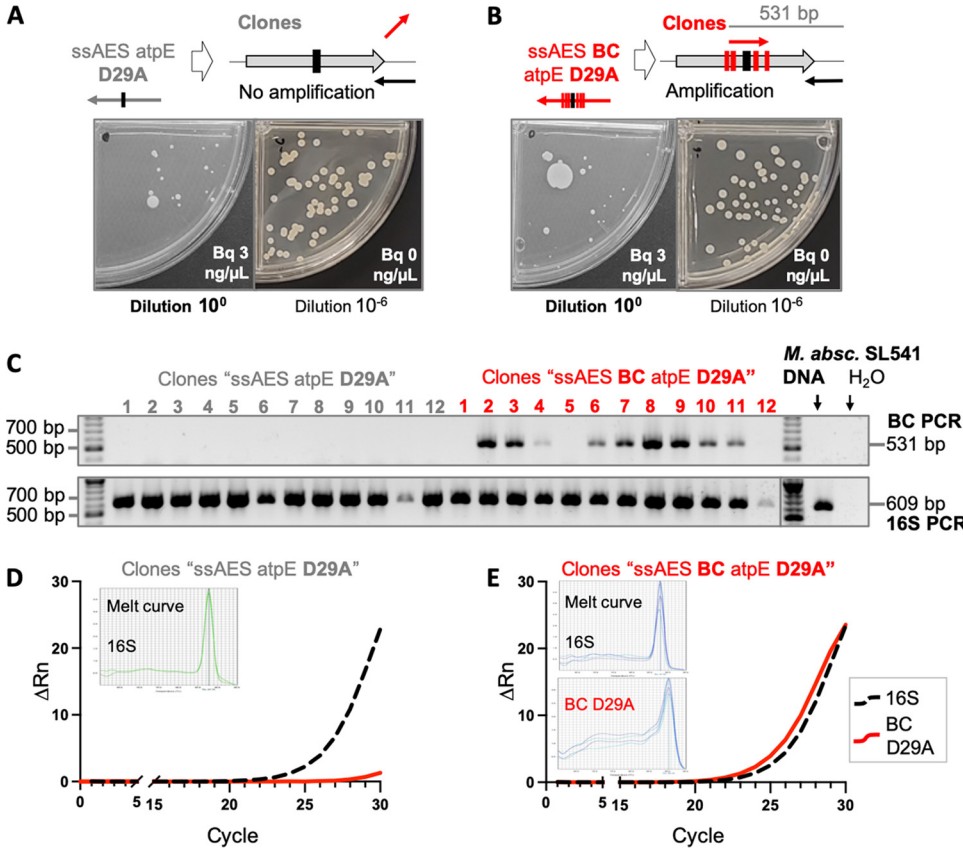

**FIG 3** Development of PCR-based methods to identify chromosomal barcodes. (A and B) Graphical representation of the nonamplification (A) and amplification (B) by barcode PCR of bedaquiline-resistant *M. abscessus* SL541 recovered after electroporation with ssAES atpE D29A and ssAES BC atpE D29A, respectively. (C) Agarose gels showing barcoding PCR amplifications of bedaquiline-resistant colonies transformed with either a nonbarcoded (gray characters) or a barcoded AES (red characters). Note the presence of the 531-bp specific band in the transformants with the barcoded AES. Amplification of a 609-bp band corresponding to the 16S housekeeping gene is also shown as a positive control for amplification. (D and E) ΔRn values of amplification of the 16S rRNA gene (dashed black) or the specific D29A barcode (continuous red) in ssAES atpE D29A (D) or ssAES BC atpE D29A (E) *M. abscessus* transformants obtained by barcoded real-time PCR.

recombineering machinery. *M. abscessus* SL541 and ATCC 19977 strains with or without the pJV53 plasmid were transformed with different AES carrying bedaquiline resistance alleles. Results showed absence of bacterial growth in the presence of bedaquiline when *M. abscessus* lacked recombineering genes. In contrast, pJV53-transformed bacteria resulted in increased CFU numbers, which were significantly higher in transformants with AES conferring bedaquiline resistance than with the wild-type *atpE* AES (Fig. S4). Together, these results demonstrated that a recombineering system is required for the chromosomal allelic exchanges reported in this study.

**The barcoding genetic strategy was proven useful to detect D29A unrelated mutations in *M. abscessus*.** Utilization of our genetic barcoding method as a general strategy to confirm antibiotic resistance mutations in *M. abscessus* required confirmation with alternative polymorphisms. Accordingly, we selected an independent drug resistance mutation in *M. abscessus*. The A64P mutation has been described in *M. abscessus* (16, 19) and also in *M. tuberculosis* (14, 16) as a polymorphism conferring bedaquiline resistance. We designed barcoded oligonucleotides carrying the 64th Pro codon flanked by synonymous substitutions as described above (Fig. 4A). Transformation of our *M. abscessus* SL541 recombineering strain with this AES and culturing transformants in bedaquiline-containing plates resulted in the growth of several drug-resistant colonies (Fig. 4B). Then, we used our PCR-based methods described above to confirm that specific incorporation of the barcode AES was responsible for

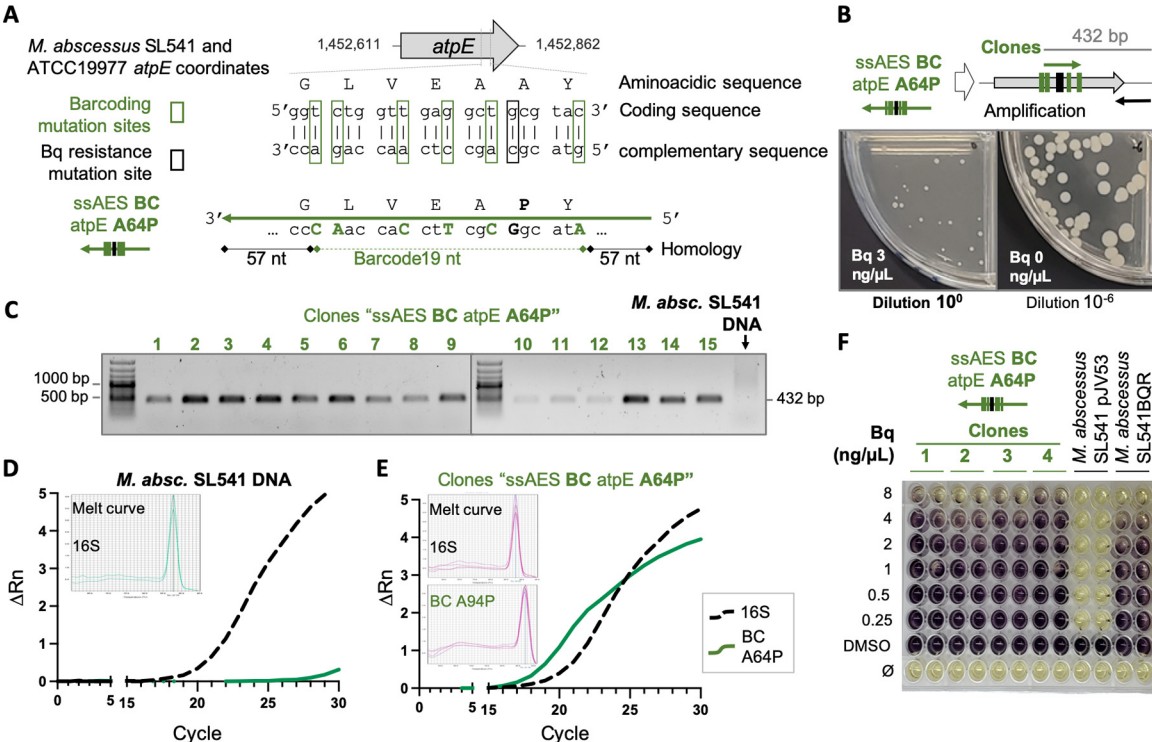

**FIG 4** Chromosomal barcoding method to establish a genotype-phenotype association for the A64P mutation in the *atpE* gene. (A) Graphical representation of the ssAES used to mutate the 64th codon of the *atpE* gene and their alignment in *M. abscessus* SL541 and reference ATTC 19977 strain genomes. (B) Graphical representation of the amplification by barcoding PCR of bedaquiline-resistant *M. abscessus* SL541 isolate recovered after electroporation with ssAES BC atpE A64P. (C) Agarose gels of barcoded PCR of bedaquiline-resistant colonies. The presence of a 432-bp band was indicative of the proper exchange of the AES shown in panel A into its specific chromosomal location. (D and E) $\Delta$Rn values of amplification of the 16S rRNA gene (dashed black) or the specific A64P barcoded (continuous green) in a WT colony (D) or ssAES BC atpE A64P *M. abscessus* SL541 transformants (E) obtained by barcode real-time PCR. (F) MTT assay results in 7H9Tyl-ADC to determine the MICs of different isolated recombinant colonies. Wells with viable bacteria are shown in dark purple, whereas wells with absence of growth are shown in yellow, after addition of MTT.

the bedaquiline resistance of the transformants. By conventional, allele-specific PCR, we confirmed PCR amplification of the 432-bp specific band in all the assayed colonies (Fig. 4C). We also confirmed specific real-time PCR amplification of the barcoded *atpE* gene in recombinant ($C_T \approx 18$) (Fig. 4E) but not in wild-type ($C_T > 31$) strains (Fig. 4D). Again, the verification process lasted from 10 to 14 days, depending on the use of real-time PCR or conventional PCR, respectively. The bedaquiline resistance of the transformants was quantitatively confirmed by the MIC, which was >8 ng/$\mu$L, in contrast to the susceptible profile with their parental *M. abscessus* recombineering strain (Fig. 4F).

## DISCUSSION

Antimicrobial resistance is a growing threat that endangers effective treatment of infectious diseases and is indeed one of the health challenges for the 21st century (17). This is particularly important for mycobacterial infections, since a limited repertoire of clinically approved drugs is available. In the era of whole-genome sequencing, exploring genome data from clinical isolates is useful to generate datasheets of putative antibiotic resistance polymorphisms. In this same context, PCR-based commercial systems to detect these polymorphisms are widely available for *M. tuberculosis* (30–32), although their introduction into nontuberculous mycobacteria is far from optimal. However, these genomic methods have previously required establishing a direct link between chromosomal polymorphisms and antimicrobial resistance, and this has been otherwise the result of long-term observation and characterization of clinical isolates.

Here, we propose a genetic method aiming to accelerate the validation of drug resistance mutations in *M. abscessus*, a bacterium which similarly to drug-resistant

*M. tuberculosis* possesses few therapeutic options (8–10). We have focused here on bedaquiline resistance mutations, since the use of this drug against *M. abscessus* infections was introduced in 2015 (33) and, accordingly, it is expected that bedaquiline resistance mutations are yet emerging after the recent introduction of the treatment. In addition, since the bedaquiline target is ATP synthase, whose *atp* coding genes show a high level of conservation between different mycobacteria (Fig. S5), it is expected that results obtained with *M. abscessus* might be able to be translated to related members of the *Mycobacterium* genus. Our method has proven robust and reliable, since it has been successfully tested with two unrelated mutations, in two independent *M. abscessus* strains, including a clinical isolate, and by independent researchers. This strategy presents several advantages with respect to other genetic methods. First, the specific chromosomal replacement avoids the use of ectopic gene expression, either in episomal or chromosomal locations. Second, the use of an available recombineering strain and the synthesis of specific oligonucleotides avoid the use of time-consuming cloning procedures. Third, results can be obtained in a reasonably short time, mostly limited to the growth of drug-resistant colonies, which usually takes more than a week. Fourth, our strategy is scalable and allows simultaneous testing of various drug resistance genotypes. These advantages might be useful to deconvolve sequencing data related to drug-resistant isolates. It is key to remember that interrogation of genome data does not always result in a list of polymorphisms located in well-known genes associated with drug resistance (34). Accordingly, in these scenarios, each individual polymorphism should be genetically evaluated for its specific contribution to drug resistance. Additionally, as this barcoding strategy allows easy identification of our mutants, it can be optimized to compare different isogenic mutants and track how mutant populations evolve upon exposure to an antibiotic pressure, resembling signature-tagged mutagenesis, a technique used to distinguish desirable mutations in pools of transposon mutants in *M. tuberculosis* (35).

It remains to be answered whether the method described here is useful to detect low-level drug resistance, since we relied on the growth of drug-resistant transformants on plates containing the corresponding drug. In this same context, further work is needed to ascertain whether other drug resistance mutations, aside those for bedaquiline, can be tested by our method. However, regarding the latter observation, we should remember that our barcoding method is suitable to establish genotype-phenotype relationships in those polymorphisms located within coding regions, because of the need to introduce silent mutations in the vicinity of the targeted polymorphisms. Accordingly, it is not always possible to introduce such silent mutations in noncoding regions, as the rRNA subunits are the targets of aminoglycosides and macrolides used in the treatment of *M. abscessus* infections. These latter cases affecting noncoding RNA targets could be otherwise examined using nonbarcoded oligonucleotides and the subsequent verification of the gene sequence. Even though the nonbarcoded method does not rule out the appearance of spontaneous mutations at the studied locus, a uniformity in the allele sequences from resistant colonies might be indicative of recombineering-derived drug-resistant colonies. Indeed, in our hands, equivalent numbers of antibiotic-resistant transformants were obtained when using nonbarcoded and barcoded AES (Fig. 2 and 3), indicating that most drug-resistant colonies arose from AES recombination and diminishing the chance of obtaining spontaneous drug-resistant mutants.

Another possible limitation of the technique relates to the high mutation frequency of *M. abscessus in vitro*. In the case of bedaquiline, we were not able to isolate significant proportions of resistant mutants in the absence of a functional recombineering system (Fig. S4). However, we cannot discard a higher frequency of spontaneous mutations when other antimicrobials are tested. Nevertheless, the introduction of barcoded mutations, and their subsequent verification using PCR-based methods, enables the discrimination between the desired mutations and false positives (Fig. 3 and 4). Another aspect to be considered is related to drugs having multiple targets or multiple modes of resistance. In this case, it is theoretically possible to introduce two or more

AES, each with specific barcode sequences, to define the individual contribution of each genotype contributing to resistance.

Another aspect to consider is that silent mutations introduced in coding regions with this barcoding strategy could alter protein expression levels due to bacterial codon usage, with the consequent change in bacterial fitness. It has been recently reported that *M. bovis* BCG is able to react to stress by tRNA reprograming and codon-biased translation. By this mechanism, BCG can drive the "over- or down-translation mRNAs" codon-biased from different families (36, 37). Taking this into account, barcoding mutations could interfere in bacterial fitness if barcoded strains were used in different experiments (like *ex vivo* and *in vivo* infection experiments). mRNA structure and hybridization with other RNA structures of the bacteria could also be affected by barcoding mutations. However, this possible problem could be minimized with a rational design of the silent mutations, by maintaining the percentage of codon usage, when feasible. On the other hand, *in silico* RNA analysis servers are improving continuously, and prediction of RNA structures and interactions could also be used to help in the design of our desired mutations.

Results described here could be theoretically applied to new candidate drugs which are currently in prelicensing phases (38). This would allow researchers to identify possible mechanisms of resistance prior to clinical evaluation of these forthcoming drugs, which might be useful to optimize diagnostic methods for the eventual drug-resistant isolates. On the other hand, since chromosomal replacements using recombineering have been described in different *Mycobacterium* species, including *M. tuberculosis*, *M. smegmatis*, *M. chelonae* (39), and *M. canettii* (unpublished results), we propose that our method could be transferable to other mycobacteria. This opens attractive possibilities to study not only drug resistance polymorphisms but also metabolic, physiological, or virulence traits in *Mycobacterium*. Overall, our genetic strategy might help to accelerate understanding the role of specific polymorphisms associated with drug resistance, with possible parallel applications to understand mycobacterial biology.

## MATERIALS AND METHODS

**Bacterial strains, culture media, and antibiotics.** We worked with clinical isolate 81327881541, identified as *M. abscessus* by matrix-assisted laser desorption ionization–time of flight analysis as described by Carbonnelle and colleagues (40) of the Cliniques Universitaires Saint-Luc, Brussels, Belgium (*M. abscessus* SL541) and the laboratory reference strain *M. abscessus* ATCC 19977 (GenBank accession number CU458896.1) (41). Bacteria were cultured at 37°C in Middlebrook 7H9 medium (Difco) supplemented with 0.05% Tween 80 (Sigma) and 10% ADC Middlebrook in 25-cm$^2$ flasks, without shaking. For plate growth, strains were incubated at 37°C in Middlebrook 7H10 (Difco) agar supplemented with 10% ADC Middlebrook. When required, kanamycin was added to *M. abscessus* cultures at 50 $\mu$g/mL to maintain the pJV53 plasmid (26) and bedaquiline was added at a maximal concentration of 8 $\mu$g/mL.

**Isolation of a spontaneous mutant resistant to bedaquiline.** A spontaneous resistant mutant to bedaquiline was obtained from a single colony that grew after plating susceptible *M. abscessus* SL541 on 7H10-ADC agar containing 4 $\mu$g/mL of bedaquiline. This colony was subcultivated to establish the *M. abscessus* SL541BQR strain, which was subjected to MIC determinations, DNA extraction, and whole-genome sequencing.

**Determination of MICs against *M. abscessus*.** MICs for *M. abscessus* were determined by the resazurin method on 7H10-ADC agar containing serial 2-fold dilutions of bedaquiline ranging from 4 to 0.25 $\mu$g/mL or kanamycin at concentrations from 50 to 3.12 $\mu$g/mL. MICs were also determined by the 3-(4,5-dimethylthiazol-2-yl)-2,5-diphenyltetrazolium bromide (MTT) method in 7H9-ADC liquid broth containing bedaquiline concentrations ranging from 8 to 0.0078 $\mu$g/mL or kanamycin ranging from 100 to 0.19 $\mu$g/mL. A total of $10^5$ CFU was added to each well in multiwell agar plates, and an initial concentration of $10^5$ CFU/mL was used to determine MICs in multiwell plates inoculated with liquid broth. Both multiwell plates were incubated at 37°C for 4 days. Then, 50 $\mu$L of 0.1-mg/mL filter-sterilized resazurin (Sigma-Aldrich) was added to agar wells, or 30 $\mu$L of 2.5-mg/mL MTT was added to liquid broth wells. MICs were defined as the lowest concentration at which no color change was observed. The Sensititre RAPMYCO susceptibility test was used, as recommended by ThermoFisher, to determine the antimicrobial resistance profile of *M. abscessus* ATCC 19977 and SL541 strains.

**Construction of recombinant mycobacteria and genetic manipulations.** To prepare electrocompetent mycobacteria, 100 mL of bacterial culture was grown to an optical density at 600 nm of 0.4 to 0.6. In the case of culturing bacteria with the recombineering system, acetamide was added to a final concentration of 0.2% (wt/vol) for approximately one doubling time of the strain used before preparing the competent cells (3 to 4 h) to allow the correct expression of the recombineering system. Bacterial pellets were washed several times in 10% glycerol–0.05% Tween 80 and finally resuspended in 1 mL of

**TABLE 2** Oligonucleotides used for recombineering AES and for PCR

| Primer name | Sequence (5′→3′)[a] | Use |
|---|---|---|
| ssAES atpE WT | gaacagccggccctgagcctcgggctgacgagccacaccggagatcagagcgttaccggcgataccgtcaccga taccggcaccgatggcgcctccggcc atgatcaacccaccaccgatgagggcaccagca | AES for recombineering |
| ssAES atpE D29A | gaacagccggccctgagcctcgggctgacgagccacaccggagatcagagcgttaccggcgataccg**G**caccg ataccggcaccgatggcgcctccggccatgatcaacccaccaccgatgagggcaccagca | AES for recombineering |
| ssAES BC atpE D29A | gaacagccggccctgagcctcgggctgacgagccacaccggagatcagagcgttacc**C**gc**A**at**C**ccg**G**c**C**cc **A**at**C**ccggcaccgatggcgcctccggccatgatcaacccaccaccgatgagggcaccagca | AES for recombineering |
| ssAES BC atpE A64P | ttagctggcgccgggagtcgcgaagacgaacaacgccatgaaggccaggttgatgaa**A**tacg**GC**gc**T**tcCac ca**AC**ccgacggtgatgaagaacggggtgaacagccggccctgagcctcgggctgacgagcc | AES for recombineering |
| Seq Mabsc atpE Fw | gccctgttcgtcttcgtctgc | PCR amplification and Sanger sequencing |
| Seq Mabsc atpE Rv | tcctcaagaatgccgcgcc | PCR amplification and Sanger sequencing |
| BC atpE D29A Fw | gattggggccgggattgcg | Barcoding PCR |
| BC atpE A64P Fw | gttggtggaagcgccgtat | Barcoding PCR |
| 16S-F | aggattagatacctggtagtcca | PCR |
| 16S-R | aggcccgggaacgtattcac | PCR |
| gp60-fw | atccggctctacgccgac | PCR |
| gp61-Rv | cggcaaatgactcttgcgt | PCR |

[a]Mismatching nucleotides introducing a single-point mutation in the AES are highlighted in bold uppercase letters.

10% glycerol. Aliquots of 100 to 200 $\mu$L were stored at −80°C for further use. The washing process was performed under cold conditions.

Aliquots of 100 to 200 $\mu$L of electrocompetent cells were electroporated with 300 to 500 $\mu$g of pJV53 plasmid DNA or 300 to 600 $\mu$g of the allelic exchange substrate (AES) for recombineering (Table 2). For selection of pJV53 transformants, kanamycin susceptibility was confirmed prior to electroporation, and incorporation of pJV53 was checked by PCR amplification of the *gp60-gp61* recombineering genes (Fig. S1). Electroporation was performed as follows: 0.2-cm gap cuvettes (Bio-Rad) were used with a single pulse (2.5 kV, 25 $\mu$F, 1,000 Ω) in a GenePulser Xcell (Bio-Rad). Cells were recovered with 5 mL of 7H9-ADC and incubated overnight at 37°C to express the antibiotic resistance genes. Serial 10-fold dilutions were plated in 7H10-ADC plates containing the relevant antibiotic. Recombinant colonies typically grew in 8 to 10 days for *M. abscessus* and were tested by colony PCR.

**PCR and DNA sequencing.** The genomic DNA from the different *M. abscessus* mutants was subjected to PCR amplification using primers listed in Table 2. Bioline MyTaq DNA polymerase was used for colony PCR amplification with the following parameters: heat denaturation at 95°C for 10 min, followed by 30 cycles of 95°C for 15 s, the corresponding annealing temperature for 15 s, and 72°C for 30 s, and then a final extension at 72°C for 2 min. PCR products were visualized in agarose gels containing ethidium bromide. PCR products for Sanger sequencing were treated with Affymetrix ExoSAP-IT PCR product cleanup (Applied Biosystems) according to the manufacturer's instructions and sequenced by STAB VIDA Corp. to confirm mutations.

For real-time colony PCR, *M. abscessus* colonies were boiled for 15 min in 20 $\mu$L of sterilized water and centrifuged at maximal speed. TaKaRa TB Green Premix *Ex Taq* was used in a Step One Plus real-time PCR system with parameters as follows: heat denaturation at 95°C for 10 min, followed by 40 cycles of 95°C for 10 s, 62°C for 10 s, and 72°C for 50 s. Primers for fast barcoded mutation detection were used at a final concentration of 0.25 $\mu$M, and DNA of *M. abscessus* colonies was added from supernatant of boiled samples. Amplification of the specific barcoded mutation was compared with amplification of the 16S rRNA gene, a positive control of amplification.

For whole-genome analysis, genomic DNA of *M. abscessus* strains under study was extracted and sequenced using Illumina MiSeq sequencing as described elsewhere (https://pubmed.ncbi.nlm.nih.gov/34751641/). Using CLC Genomics Workbench v20.0 (Qiagen), trimmed reads were either mapped against *M. abscessus* ATCC 19977 or *de novo* assembled. Variant calls for genes of interest were performed using CLC's variant caller at high stringency, with minimal position coverage of 30× and minimal frequency of 90%.

**Ethics statement.** We declare that, once the manuscript is published, all drug-resistant strains generated in this work will be properly eliminated.

**Data availability.** Data required to reproduce the content of the manuscript have been described elsewhere. In case of additional information, data will be provided upon request to the corresponding author.

## SUPPLEMENTAL MATERIAL

Supplemental material is available online only.
**SUPPLEMENTAL FILE 1**, PDF file, 8.1 MB.

## ACKNOWLEDGMENTS

Designed the study, J.C.-S., A.M., and J.G.-A.; performed the experiments, J.C.-S., E.C.-Y., V.M., and P.-J.C.; analyzed the data, J.C.-S., E.C.-Y., V.M., P.-J.C., and J.G.-A.; provided

biological material, H.R.-V. and A.M.; conceived figures, J.C.-S. and J.G.-A.; designed figures, J.C.-S.; wrote the original draft, J.C.-S. and J.G.-A.; reviewed the final version of the manuscript, J.C.-S. and J.G.-A.; obtained funding, J.G.-A.

This work was supported by grants from "Gobierno de Aragón-Fondo Europeo de Desarrollo Regional (FEDER) 2014-2020: Construyendo Europa Desde Aragón" to J.C.-S., grant PRE2020-096507 funded by MCIN/AEI 10.13039/501100011033 to E.C.-Y., and by grant PID2019-104690RB-I00 funded by MCIN/AEI/10.13039/501100011033 to J.G.-A.

We declare we have no known competing financial or personal interest relationships.

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
