## [Reviewer comments · Microbiology Spectrum]

Microbiology Spectrum

Targeted chromosomal barcoding establishes direct genotype-phenotype associations for antibiotic resistance in *Mycobacterium abscessus*

Juan Galvet Seral, Estefanía Crespo-Yuste, Vanessa Mathys, Hector Rodriguez-Villalobos, Pieter-Jan Ceyskens, Anandi Martin, and Jesus Gonzalo-Asensio

Corresponding Author(s): Jesus Gonzalo-Asensio, Universidad de Zaragoza Departamento de Microbiologia Medicina Preventiva y Salud Publica

Review Timeline:

Submission Date:	December 28, 2022
Editorial Decision:	January 20, 2023
Revision Received:	February 24, 2023
Accepted:	March 4, 2023

Editor: Olivier Neyrolles

Reviewer(s): The reviewers have opted to remain anonymous.

Transaction Report:

DOI: <https://doi.org/10.1128/spectrum.05344-22>

January 20, 2023

Dr. Jesus Gonzalo-Asensio
Universidad de Zaragoza Departamento de Microbiologia Medicina Preventiva y Salud Publica
Zaragoza, Zaragoza 50009
Spain

Re: Spectrum05344-22 (Targeted chromosomal barcoding establishes direct genotype-phenotype associations for antibiotic resistance in *Mycobacterium abscessus*)

Dear Dr. Jesus Gonzalo-Asensio:

Link Not Available

Sincerely,

Olivier Neyrolles
Editor, Microbiology Spectrum

Journals Department
Reviewer comments:

Reviewer #1 (Comments for the Author):

Bedaquiline (BDQ) has recently been shown to be active against *M. abscessus*. In this study, Calvet-Seral et al. developed a genetic strategy to establish a direct genotype-phenotype association for resistance to BDQ in *M. abscessus*. This method is based on the recombineering strategy and using a genetic barcode of silent mutations in the codons flanking the D29A and A64P mutations (identified earlier as key mutations in resistance to BDQ in *M. abscessus*). The authors clearly showed that the genetic barcode was associated to the target incorporation of the D29A and A64P mutations, in both the reference strain ATCC19977 as well as the clinical strain SL541. From a biological perspective, this work only confirms already characterized mutations involved in BDQ resistance in *M. abscessus* and does not provide new insights into the mechanisms of drug resistance. The major take-home message relies on the successful demonstration of the use of barcoded oligo-mediated targeted mutagenesis in the genome of *M. abscessus*, although the use of the recombineering strategy is not innovative as it has been widely used in different mycobacterial species, including *M. abscessus*. In general, the manuscript is very well written

and the conclusions raised are supported by the data. However, there are a few points/controls that need to be clarified to improve the quality of the manuscript.

Major points:

-line 151: the authors mention that the pJV53 was maintained in the presence of 50µg/ml kanamycin. Usually, the MIC of kanamycin is much higher than 50µg/ml in *M. abscessus* and very high concentrations are needed for the selection of a plasmid. Selection on low concentrations is often associated to false-positive clones growing on kanamycin but without containing the plasmid of interest. Therefore, I am not convinced that the *M. abscessus* strains used in this study contain the pJV53. This needs to be checked either by PCR and or Western blotting using anti-gp61 antibodies. Since *M. abscessus* is able to perform homologous recombination, it is still possible the double recombination event required for the allelic exchange occurred even in the absence of pJV53. This needs to be clarified experimentally.

-the authors indicate at several places in the manuscript (as well as in Figure 1), that results can be obtained in a reasonable short time, mostly limited to the growth of drug resistant colonies. While this is probably true, they did not mention that prior to electroporation of the single stranded allelic exchange substrates, the strain containing pJV53 needs to be generated first, which can be challenging in the case of a clinical strain as mentioned above. This may add an extra 2 weeks to the overall timescale, and should be highlighted in the text as well as in Figure 1.

-provide the MICs of the ATCC strain in Table 2, so that the reader can compare the drug susceptibility profiles of the two strains used in this study. Indicate also the MIC of BDQ and kanamycin for both strains.

Minor points:

-The section in the Discussion describing drug resistance mechanisms in *M. tuberculosis* is out of the scope of this study.

Therefore, I propose the authors to delete this section (lines 370 to 377).

-line 150 "Kanamycin" instead of "Kanamicyn"

-line 229: "establish" instead of "stablish"

-line 297: "flanked" instead of "flaked"

Reviewer #2 (Comments for the Author):

Drug resistance is a growing problem in the control of mycobacterial infections. Various strategies have been used to establish a functional link between a SNP and the resistance to a drug. Most available methods have limitations that are mostly associated with unwanted side effects of genetic complementation of a SNP either by chromosomal exchange or episomal introduction of a wt copy of the resistance gene. To overcome these disadvantages, the authors have employed an elegant recombineering system developed by the Hartfull lab, that enables chromosomal point mutagenesis. Using a BDQ resistant clinical isolate of *M. abscessus*, they identified a SNP in the *atpE* gene, which is used to showcase their methodology. In addition, the authors use "chromosomal barcoding" (=silent SNPs near the resistance conferring mutation) to distinguish intended mutagenesis from random (natural) mutagenesis events. Performance of the new genotype-phenotype association method introduced here was verified by Sanger sequencing, allele specific PCR, qPCR and drug susceptibility testing of mutant *M. abscessus* colonies. Experimental data shown is of good scientific rigor and supports the conclusions made. Experiments were carefully designed and executed. The mostly well-written manuscript has very minor deficiencies in language - some are included below - others will be easily addressed by the copyeditors. The method of genotype-phenotype association is of interest to the field of drug development/resistance and certainly to readers of *Spectrum* as well. The discussion could be condensed, e.g. it is not overly important for the presented data to discuss *M. tuberculosis* drug resistance in detail (1st paragraph). The focus of the discussion should be the limitations (see below) and applications of the new methodology introduced by the authors. The points below should be considered for improvement of this overall exciting work.

Major points:

1. Fig. 3C and S2: negative PCR reactions are in general less appreciated. Amplification of a housekeeping gene should be included to verify that the absence of a band is not the result of a non-working PCR reaction.

Fig. 3C: Why did two of the clones shown in the right panel fail to produce the expected amplicon? Did the PCR reaction not work, or have they gained BDQ resistance through mechanisms not involving *atpE* D29A? This needs to be explained.

2. The Discussion could be condensed, and further possible limitations of the introduced method should be included: (1) Impact of high in vitro *M. abscessus* mutation frequency of a study drug on the application of the new methodology. (2) How useful is the author's methodology to study drugs having multiple targets or multiple modes of resistance?

Minor points:

1. L127: "circularize lineal dsDNA" should read "circularize linear dsDNA".

2. L155: "SL541 in 7H10-ADC" should read "SL541 on 7H10-ADC agar".

3. L156: "to stalish" should read "to establish".
4. L162: "plates were incubated": do the authors refer here to agar plates or multi-well plates with liquid cultures? This should be specified.
5. L165: "to broth plates": to avoid confusion, the authors may consider to further specify such as "to broth cultures in multi-well plates".
6. L190-191: three instances "seg" should read "sec".
7. L198-199: three instances "seg" should read "sec".
8. L202: "16s gene" should read "16S rRNA".
9. L229: "to stalish" should read "to establish".
10. L236: It is not entirely intuitive to reference Koch's postulates (published in the pre-antibiotics era) in the context of drug resistance. The authors may consider removing this to avoid confusion to some readers.
11. L250: "selected in antibiotic containing plates" should read "selected on antibiotic containing plates".
12. L255: "in 7H10-ADC agar" should read "on 7H10-ADC agar".
13. L667: "16S RNA" should read "16S rRNA".
14. Fig. 1E: "16s" should read "16S".

Staff Comments:

Preparing Revision Guidelines

Please return the manuscript within 60 days; if you cannot complete the modification within this time period, please contact me. If you do not wish to modify the manuscript and prefer to submit it to another journal, please notify me of your decision immediately so that the manuscript may be formally withdrawn from consideration by Microbiology Spectrum.

Manuscript reference number: **Spectrum05344-22**

Manuscript title: **Targeted chromosomal barcoding strategy allows establishing direct genotype-phenotype associations for antibiotic resistance in *Mycobacterium abscessus***

Zaragoza 21th february, 2023

Dear Dr. Neyrolles,

Please find enclosed a substantially **revised version of the manuscript** "Targeted chromosomal barcoding strategy allows establishing direct genotype-phenotype associations for antibiotic resistance in *Mycobacterium abscessus*" by Calvet-Seral and cols. We thank you for the editorial and reviewers' comments on our recent submission to Microbiology Spectrum. Please find **each comment in blue below, followed directly by the authors' response**. Any changes made to the manuscript itself have been highlighted in yellow. The revised manuscript contains **2 updated Figure panels, 2 additional Supplementary Figures and a new Table**, aside from the requested text changes. Overall, we consider that the revised manuscript has significantly improved its quality after the reviewer's recommendations and we hope that it is suitable for publication in Microbiology Spectrum.

Yours faithfully,

Jesús Gonzalo-Asensio, PhD; on behalf of the co-authors of this work

Reviewer #1 (Comments for the Author):

We acknowledge the overall positive impression of the reviewer. Please find below a point-by-point answers to your comments which has also resulted in the inclusion of new figures/tables in this revised manuscript.

Major points:

-line 151: the authors mention that the pJV53 was maintained in the presence of 50µg/ml kanamycin. Usually, the MIC of kanamycin is much higher than 50µg/ml in *M. abscessus* and very high concentrations are needed for the selection of a plasmid. Selection on low concentrations is often associated to false-positive clones growing on kanamycin but without containing the plasmid of interest. Therefore, I am not convinced that the *abscessus* strains used in this study contain the pJV53. This needs to be checked either by PCR and or Western blotting using anti-gp61 antibodies.

We thank the reviewer for noticing this discrepancy, which in our opinion comes from the various kanamycin concentrations used in the literature to select *M. abscessus* recombinants. While some studies use high kanamycin concentrations (i. e. 250 µg/mL), other studies use lower concentrations (in the range of 25-100 µg/mL) of this antibiotic to select either mutant, or plasmid-carrying, bacteria.

Since we routinely work in mycobacterial genetics, we are aware of the diverse levels of antibiotic susceptibility displayed by laboratory and clinical strains. Accordingly, prior to introduce pJV53 into *M. abscessus*, we checked the antibiotic susceptibility against kanamycin in both, the ATCC19977, and the SL541, strains of *M. abscessus*.

The picture above shows MICs against kanamycin and gentamycin of the aforementioned strains, showing that MIC against kanamycin lies between 12.5 and 6.25 $\mu\text{g/mL}$ in the SL541 strain, and between 3.12 and 1.56 $\mu\text{g/mL}$ in the ATCC19977 strain. Thus, both *M. abscessus* strains used in this study are susceptible to kanamycin at 50 $\mu\text{g/mL}$.

Therefore, selection of *M. abscessus* pJV53 transformants on plates containing 50 $\mu\text{g/mL}$ results in an efficient recovery of colonies as shown in the picture above. Note that non-transformed bacteria are unable to grow on 50 $\mu\text{g/mL}$ kanamycin plates, which minimizes the chance of recovering spontaneous mutants against this antibiotic.

Nevertheless, to molecularly confirm the presence of pJV53 into *M. abscessus* colonies grown on 50 $\mu\text{g/mL}$ kanamycin plates, we checked the presence of the plasmid by PCR amplification of the *gp60-gp61* recombining genes contained in the plasmid. Results confirmed the presence of pJV53 in most colonies analyzed. We carefully selected pJV53-bearing colonies for our downstream experiments.

Finally, prior to perform recombineering experiments, we checked again the presence of pJV53 into *M. abscessus* by assessing the kanamycin resistance profile of our recombineering strains. As shown above, an SL541 strain carrying pJV53 exhibits a kanamycin resistance profile, in contrast to the wild type SL541, or the bedaquiline resistant mutant of this strain.

All these results are included as a new Supplementary Figure 1, since we consider that they add transparency and reproducibility to the methodology. In addition, related paragraphs have been added to the methods and results section. (lines 182-185, 261-264, 337-339)

Since *M. abscessus* is able to perform homologous recombination, it is still possible the double recombination event required for the allelic exchange occurred even in the absence of recombination, it is still possible the double recombination event required for the allelic exchange occurred even in the absence of pJV53. This needs to be clarified experimentally.

This is a very relevant observation which we overlooked in the initial submission. We repeated the experimental flow described in the original manuscript with *M. abscessus* SL541 and ATCC19977 strains, with or without pJV53. All strains were grown in parallel, and 600 ng of each AES (*atpE* wt as negative control, *atpE* D29A with or without barcode, and *atpE* A64P with barcode), were electroporated into 200 μ L of electrocompetent cells. 100 μ L of transformed bacteria were plated at the indicated dilution in presence or absence of Bedaquiline. Results showed absence of bacterial growth in wild type bacteria not transformed with pJV53. In contrast, pJV53-transformed bacteria resulted in increased CFU numbers, which were significantly higher in transformants with AES conferring bedaquiline resistance (*atpE* D29A with or without barcode, and *atpE* A64P with barcode), when compared with the *atpE* wt AES. Differences in CFU numbers are not related to differences in bacteria plated as observed in the CFU grown in the absence of bedaquiline.

Together, we consider that these results unequivocally demonstrate that a recombinering system is required for the allelic exchanges reported in the manuscript, and we sincerely acknowledge this reviewer's clarification. A new Supplementary Figure 4 including these results in the SL541 and ATCC19977 strains is provided, and additional sentences clarifying this suggestion have been included in the manuscript (lines 350-359, 754-763)

-the authors indicate at several places in the manuscript (as well as in Figure 1), that results can be obtain in a reasonable short time, mostly limited to the growth of drug resistant colonies. While this is probably true, they did not mention that prior to electroporation of the single stranded allelic exchange substrates, the strain containing pJV53 needs to be generated first, which can be challenging in the case of a clinical strain as mentioned above. This may add an extra 2 weeks to the overall timescale, and should be highlighted in the text as well as in Figure 1.

The reviewer is right. A new panel in Figure 1 has been added to indicate the experimental flow and timelines for the construction of a recombinering strain, if needed. The challenge to transform non-laboratory strains is also indicated elsewhere (lines 337-339, 675-677)

-provide the MICs of the ATCC strain in Table 2, so that the reader can compare the drug susceptibility profiles of the two strains used in this study. Indicate also the MIC of BDQ and kanamycin for both strains.

This is an important consideration for potential readers. Table 2 has been updated with data from the ATCC19977 strains, and including bedaquiline and kanamycin MICs for both strains (lines 664-666)

Minor points:

-The section in the Discussion describing drug resistance mechanisms in *M. tuberculosis* is out of the scope of this study. Therefore, I propose the authors to delete this section (lines 370 to 377).

We agree with the reviewer, and this paragraph has been deleted to provide a more focused message.

-line 150 "Kanamycin" instead of "Kanamicyn"

Corrected, line 149

-line 229: "establish" instead of "stablish"

Corrected, line 234

-line 297: "flanked" instead of "flaked"

Corrected, line 302

Reviewer #2 (Comments for the Author):

We thank the reviewer for the encouraging comments and we hope to have addressed her/his suggestions in the revised version of the text, and figures.

Major points:

1. Fig. 3C and S2: negative PCR reactions are in general less appreciated. Amplification of a housekeeping gene should be included to verify that the absence of a band is not the result of a non-working PCR reaction.

The reviewer is right and we greatly acknowledge his/her suggestion since it contributes to improve the overall reliability of the results. We repeated the PCR using DNA from the original SL541 colonies transformed with either the barcode or the non-barcode D29A oligonucleotide. In parallel, we amplified by PCR the coding region of the 16S rRNA housekeeping gene, using the same oligonucleotides used in the qRT-PCR reactions shown in Figure 3D. Results show amplification of the 16S gene in all colonies analyzed, irrespective of their transformation with barcoded or non-barcoded oligonucleotide. Concerning amplification of the barcode region, this time we obtained amplification in 9/12 colonies, compared to the positive amplification of 10/12 colonies in the original submission. Colonies 1 and 5 failed to produce amplification in both cases. Colony 12 showed a slight PCR amplification in our original submission, but failed to produce a PCR band in this revised version. However, observing the PCR amplification of the 16S control gene in this colony, it is evident that the low amounts of template DNA might influence the outcome of the PCR reaction in this case. This same approach was performed with ATCC19977 D29A bedaquiline resistant colonies in the new Figure S3 (formerly S2). Together, results were highly reproducible, and the inclusion of an unrelated, positive control of amplification is useful to discriminate putative false positives due to spontaneous mutations. Panel C of Figure 3, and Figure S3 (formerly S2), have been updated and a description of the positive control is provided in lines 319-321, 699-700, and 750-751.

Fig. 3C: Why did two of the clones shown in the right panel fail to produce the expected amplicon? Did the PCR reaction not work, or have they gained BDQ resistance through mechanisms not involving *atpE* D29A? This needs to be explained.

As explained in the previous point, the reviewer's suggestion to include an unrelated, positive control of PCR amplification is useful to discriminate between the desired mutations and false positives due to spontaneous mutations also conferring bedaquiline resistance. Thus, colonies 1 and 5 probably arose from a spontaneous mutation either in the *atpE* gene or elsewhere in the chromosome. Colony 12, as shown above, is a true D29A mutation selected by recombineering, albeit the low quantities of template DNA do not allow PCR amplification in this case. For the sake of clarity, and to provide a clear and straightforward message to the reader, these 3 colonies have been considered as false positives, and these points are summarized in lines 321-325.

2. The Discussion could be condensed, and further possible limitations of the introduced method should be included:
 - (1) Impact of high *in vitro* *M. abscessus* mutation frequency of a study drug on the application of the new methodology.
 - (2) How useful is the author's methodology to study drugs having multiple targets or multiple modes of resistance?

In line with the suggestions of both reviewers, some paragraphs, including those describing antimicrobial resistance in *M. tuberculosis*, have been eliminated. These additional limitations of the technique are discussed in lines 448-458.

Minor points:

We acknowledge the reviewers for spotting these typographic errors

1. L127: "circularize lineal dsDNA" should read "circularize linear dsDNA".

Corrected, line 126

2. L155: "SL541 in 7H10-ADC" should read "SL541 on 7H10-ADC agar".

Corrected, line 154

3. L156: "to stalish" should read "to establish".

Corrected, line 155

4. L162: "plates were incubated": do the authors refer here to agar plates or multi-well plates with liquid cultures? This should be specified.

The reviewer is right. This methodology is confusing and not correctly explained in the initial submission, and we acknowledge this constructive criticism. We used multi-well cultures to quantify the MIC. These multi-well plates were incubated in either solid or liquid media to enumerate MICs by the resazurin, and the MTT methods, respectively. The complete method section has been rewritten (lines 157-168)

5. L165: "to broth plates": to avoid confusion, the authors may consider to further specify such as "to broth cultures in multi-well plates".

Corrected in the context of the previous comment, lines 166-167

6. L190-191: three instances "seg" should read "sec".

Corrected, lines 194-195

7. L198-199: three instances "seg" should read "sec".

Corrected, lines 203-204

8. L202: "16s gene" should read "16S rRNA".

Corrected, line 207

9. L229: "to stalish" should read "to establish".

Corrected, line 234

10. L236: It is not entirely intuitive to reference Koch's postulates (published in the pre-antibiotics era) in the context of drug resistance. The authors may consider removing this to avoid confusion to some readers.

Amended, lines 241-242

11. L250: "selected in antibiotic containing plates" should read "selected on antibiotic containing plates".

Corrected, line 253

12. L255: "in 7H10-ADC agar" should read "on 7H10-ADC agar".

Corrected, line 258

13. L667: "16S RNA" should read "16S rRNA".

Corrected, line 701

14. Fig. 1E: "16s" should read "16S".

Corrected in the updated Figure 1

March 4, 2023

Dr. Jesus Gonzalo-Asensio
Universidad de Zaragoza Departamento de Microbiologia Medicina Preventiva y Salud Publica
Zaragoza, Zaragoza 50009
Spain

Re: Spectrum05344-22R1 (Targeted chromosomal barcoding establishes direct genotype-phenotype associations for antibiotic resistance in *Mycobacterium abscessus*)

Dear Dr. Jesus Gonzalo-Asensio:

Congratulations! Your manuscript has been accepted, and I am forwarding it to the ASM Journals Department for publication. You will be notified when your proofs are ready to be viewed.

Sincerely,

Olivier Neyrolles
Editor, Microbiology Spectrum
